# What treatments work for anxiety and depression in children and adolescents with chronic fatigue syndrome? An updated systematic review

Philippa Clery ,[1] Alexander Royston,[1] Katie Driver,[1] Jasmine Bailey,[1] Esther Crawley,[1,2] Maria Loades[1,3]

[1]Centre for Academic Child Health, University of Bristol, Bristol, UK
[2]Paediatric Chronic Fatigue Syndrome Specialist Service, Royal United Hospitals Bath NHS Foundation Trust, Bath, UK
[3]Department of Psychology, University of Bath, Bath, UK

**Correspondence to**
Dr Philippa Clery;
philippa.clery@bristol.ac.uk

## ABSTRACT

**Objectives** Children with chronic fatigue syndrome/myalgic encephalomyelitis (CFS/ME) experience a higher prevalence of depression and anxiety compared with age-matched controls. Our previous systematic reviews in 2015/16 found little evidence for effective treatment for children with CFS/ME with comorbid depression and/or anxiety. This review updates these findings.

**Design** A systematic review. We searched Cochrane library, Medline, Embase and PsycINFO databases from 2015 to 2020. We combined the updated results with our previous reviews in a narrative synthesis.

**Participants** Inclusion criteria: <18 years old; diagnosed with CFS/ME (using Centers for Disease Control and Prevention, National Institute for Health and Care Excellence or Oxford criteria); validated measures of depression and/or anxiety.

**Interventions** Observational studies or randomised controlled trials.

**Comparison** Any or none.

**Outcomes** Studies with outcome measures of anxiety, depression or fatigue.

**Results** The updated review identified two studies. This brings the total number of paediatric CFS/ME studies with a measure of anxiety and/or depression since 1991 to 16. None of the studies specifically targeted depression, nor anxiety. One new study showed the Lightning Process (in addition to specialist care) was more effective at reducing depressive and anxiety symptoms compared with specialist care alone. Previous studies evaluated cognitive–behavioural therapy (CBT); pharmacological interventions and behavioural approaches. CBT-type interventions had most evidence for improving comorbid anxiety and/or depressive symptoms but varied in delivery and modality. Other interventions showed promise but studies were small and have not been replicated.

**Conclusion** Very few paediatric CFS/ME intervention studies have been conducted. This review update does not significantly add to what is known from previous reviews. The evidence is of poor quality and insufficient to conclude which interventions are effective at treating comorbid anxiety and/or depression in paediatric CFS/ME.

**PROSPERO registration numbers** CRD42016043488 and CRD42015016813.

### Strengths and limitations of this study

► This review used a systematic approach to identify updated evidence for treatment approaches for comorbid anxiety and/or depression in paediatric chronic fatigue syndrome/myalgic encephalomyelitis, and combined it with previous review results to provide a comprehensive synthesis of all evidence available since 1991.

► Non-English language articles were included.

► Authors were contacted and subgroup data obtained when available.

► Grey literature and unpublished material was not included.

► There was insufficient data to carry out a meta-analysis.

## INTRODUCTION

Chronic fatigue syndrome (CFS)/ myalgic encephalomyelitis (ME) is a common but poorly understood condition causing disabling fatigue, malaise, myalgia, sleep difficulties and problems concentrating.[1] In children and adolescents (henceforth referred to as children), prevalence is estimated at 0.55% (95% CI 0.22% to 1.35%) across community, primary care and hospital populations.[2] CFS/ME has long-term impacts on children's physical, cognitive, emotional and social functioning.[3 4]

Children with CFS/ME suffer from higher rates of both depression and anxiety than age-matched population samples. The prevalence estimates of comorbid depression and anxiety are 20%[5] and 29%,[6] respectively, compared with 2.1% and 7.2%[7] in adolescents without CFS/ME. In those attending a specialist CFS/ME service, 61% who meet diagnostic criteria for depression also have an anxiety disorder.[5] Having comorbid depression and/or anxiety is associated with less favourable outcomes and may impact on engaging with treatment.

Comorbid depression in paediatric CFS/ME is associated with greater functional disability, worse fatigue and more pain compared with those without depression.[8 9] Low mood, anergia and anhedonia could be barriers to motivation to engage in behavioural treatment approaches and cognitive behavioural therapy for fatigue (CBT-f). Depressive symptoms are therefore likely to require tailored treatment.[9] The impact of anxiety on outcomes is less clear. Given that most children with CFS/ME who have anxiety also have depression,[5] it is important to explore treatments for both.

Despite the high prevalence of comorbid mental health problems, there is little evidence about the effectiveness of treatments. Our two previous systematic reviews looking at depression and anxiety outcomes in existing CFS/ME intervention studies found that no specifically adapted treatments had been trialled to target depression and anxiety in paediatric CFS/ME.[10 11] Although CBT-f and a multicomponent inpatient programme showed promise in reducing depressive[10] and anxiety[11] symptoms, there was no consistent treatment approach for children with CFS/ME and comorbid depression or anxiety. Since conducting these reviews in 2015/2016, further intervention studies may have been published. It is important and timely to review the current evidence to provide an update on what treatments should be offered to this population. Further, it is important to consider anxiety and depression together given their overlap, whereas our previous reviews considered them separately.

We conducted an updated systematic review by synthesising the evidence regarding treatments for paediatric CFS/ME and comorbid depression and anxiety since 2015. We combined these findings with results from our previous systematic reviews (1991–2015) to give an overview of all interventions evaluated since 1991 (when CFS/ME was scientifically defined). Specifically, we aimed to address the following:

1. What treatment approaches are there for depression and anxiety in children with CFS/ME?
2. What is known about the treatment efficacy of these approaches for treating depression and anxiety in CFS/ME? Do different approaches have different outcomes?

## METHODS
### Data sources and search strategy
We conducted searches on Medline, Embase, PsycINFO and Cochrane Library databases. Searches were designed with input from an information specialist to include the concepts: paediatric; CFS/ME; anxiety and depression (search strategies are in online supplemental material). We updated the searches from when they had last been run (February 2015 for depression search; July 2016 for anxiety search) up until September 2020. The two searches were carried out by different reviewer teams: anxiety search (PC and AR); depression search (KD and

**Table 1**  Inclusion criteria

| | Anxiety review | Depression review |
|---|---|---|
| Participants | 1. Children <18 years of age 2. Diagnosed with CFS/ME defined using one of these criteria: CDC aka Fukuda *et al*[50] NICE[1] Oxford aka Sharpe *et al*[51] | |
| Interventions | Observational cohort studies Any study with intervention—for example, observational clinical cohorts, clinical trials. | |
| Baseline measure | Validated assessment of anxiety | Validated assessment of depression |
| Outcome measure | Either an anxiety and/or fatigue measure on psychometrically validated assessments or validated diagnostic interviews. | Either a depression and/or fatigue measure on psychometrically validated assessments or validated diagnostic interviews. |
| Language | Non-English language papers were considered for inclusion. | |

CDC, Centers for Disease Control and Prevention; CFS/ME, chronic fatigue syndrome/myalgic encephalomyelitis; NICE, National Institute for Health and Care Excellence.

JB). Grey literature was not searched. Reference lists of articles for full-text screening were hand-searched.

### Inclusion and exclusion criteria
Studies were included if they met inclusion criteria (table 1).

### Study selection
Articles returned from database searches were inputted into Endnote and duplicates removed. Each reviewer (PC, AR, KD, JB) conducted title and abstract screening independently. Full texts of potentially eligible articles were screened against specifically created eligibility checklists. The final articles for inclusion were cross-checked between all four reviewers and any conflicts discussed and resolved with input from the senior author (ML) if necessary. Where information from the paper was insufficient to determine eligibility, authors were contacted by email for additional information. If authors did not reply after two follow-up emails, the study was excluded. Figure 1 presents the PRISMA[12] flow chart.

### Data extraction
For all included articles, data were extracted independently by two reviewers (PC and AR) using a purpose-designed data extraction form to collect information about: study design; setting; recruitment; participant characteristics; CFS/ME definition used for diagnosis; assessment of depression and anxiety; other outcomes; treatment and interventions provided; definition of response and treatment/intervention outcomes.

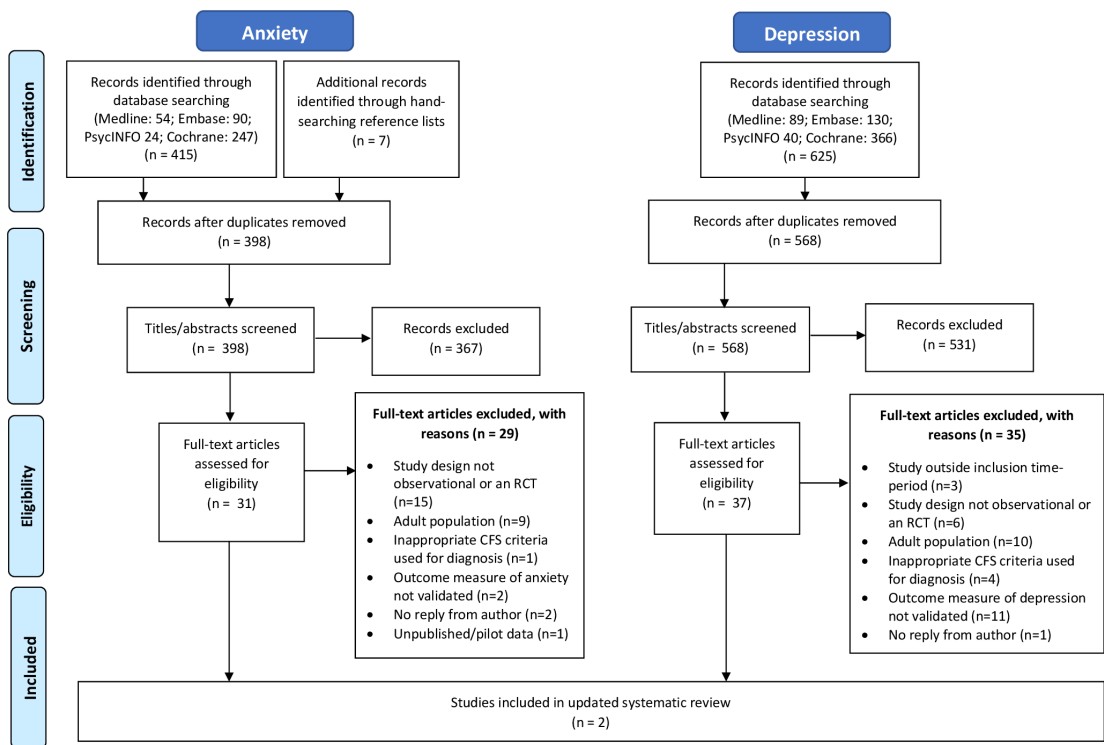

**Figure 1** Flow chart for studies included in the systematic review; based on PRISMA guidelines. CFS, chronic fatigue syndrome; PRISMA, Preferred Reporting Items for Systematic Reviews and Meta-Analyses; RCT, randomised controlled trials.

## Quality assessment

PC and AR used Risk of Bias (ROB) assessment tools[13 14] to assess methodological quality of the included studies.

## Data synthesis

We combined results from the included studies identified in the updated search with findings from the two previous systematic reviews[10 11] to conduct a narrative synthesis,[15] providing an overview of all longitudinal studies that have been evaluated in this clinical cohort since 1991 (when CFS/ME was scientifically defined). There was insufficient comparable data to conduct a meta-analysis as interventions were heterogeneous and a range of outcome measures were reported. For each of the new studies, the effects of interventions on outcomes using mean differences were compared.

## Patient and public involvement

No patients were involved.

## RESULTS
## Studies included

In the updated search (2015–2020), a total of 625 and 415 references were found by database searching for the depression and anxiety searches, respectively. After full-text screening, both searches returned the same two eligible studies.[16 17] One was a randomised controlled trial (RCT),[17] one was a retrospective observational cohort study.[16] The PRISMA[12] flow chart is in figure 1.

The previous systematic reviews for depression[10] (search conducted in 2015) and anxiety[11] (search conducted

in 2016) found 362 and 1274 references, respectively. After full-text screening, the depression search returned nine eligible studies (one RCT,[18] and eight observational studies[19–26]), and the anxiety search returned nine eligible papers from eight studies (three RCTs,[27–30] six observational studies[19 21 22 25 31 32]). Four of the studies from these two searches were the same.

Therefore, in total, 16 eligible studies were included in this narrative synthesis review. Figure 2 shows a flow chart combining studies from this updated search with studies identified from previous reviews.

## Quality assessment

Of the total 16 studies in this review, 10 were observational and six were RCTs. Of the observational studies, five had an overall ROB as 'unclear', and five had 'high' ROB (as defined by the Cochrane ROB scale, ROBINS-I (Risk of Bias In Non-Randomised Studies)[13]). Of the RCTs, all six had an overall rating of 'low' ROB (as defined by the Cochrane ROB-2 scale).[14] See online supplemental material for the quality assessment table. For detailed reporting on the quality assessment of studies from the previous searches, please refer to our previous two reviews.[10 11]

In this paper, we report in detail on the quality assessment of the two new studies found in the updated search.

The RCT[17] was conducted by members of our CFS/ME research team (EC). The study has a low ROB from the concealed allocation randomisation process, minimal deviation from how interventions were intended to be delivered, and appropriate intention-to-treat analysis. Outcome measurement is biased because of self-reported

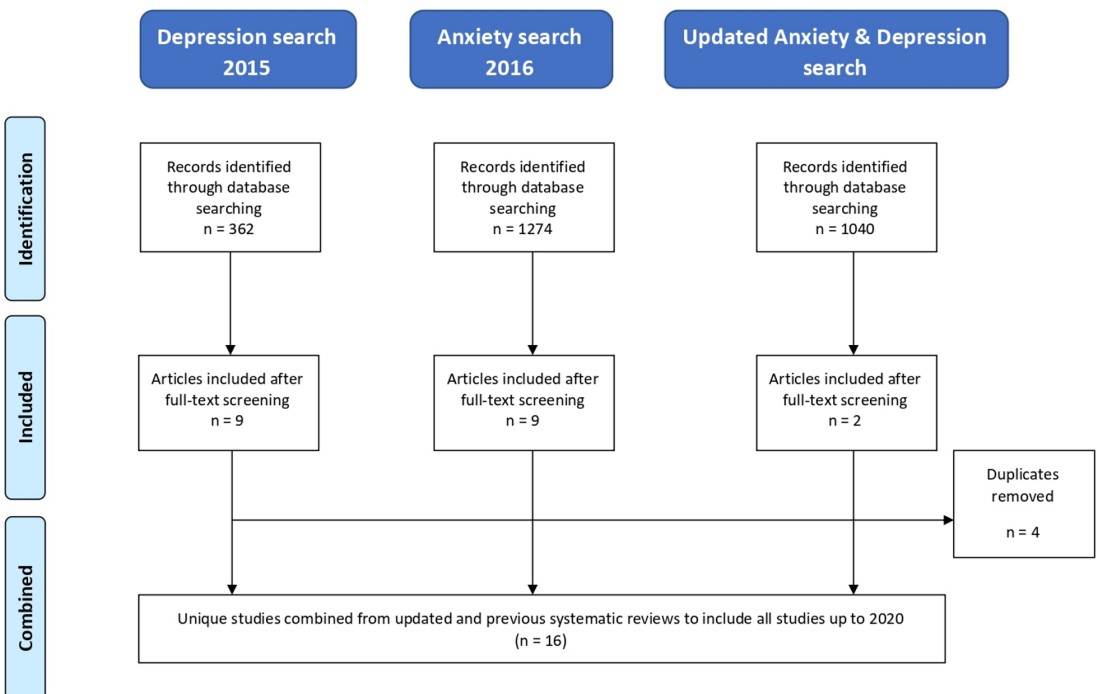

**Figure 2** Flow chart of studies combined from updated review and previous reviews.

measures, but this is standard for behavioural treatments. It is also biased due to loss to follow-up. In the control arm at 3 months, 13 of 49 (27%) were lost to follow-up and at the primary outcome of 6 months, 12 of 49 (24%) were not included in analysis. In the intervention arm 8 of 51 (16%) were lost to follow-up at 3 months and 7 of 51 (14%) were not included in primary analysis at 6 months. Although baseline characteristics between those who did and did not provide primary outcome data were similar, it is possible that missingness was related to the outcome.

The retrospective observational study[16] is also biased due to poor follow-up rates at any one time point (making comparison difficult), and no pre-published analysis plan. In the cohort, there are two samples; one with baseline data for anxiety and depression and one without. Follow-up questionnaires were mailed to all participants on a number of occasions between January 2008 and June 2011. This produced a range of follow-up time points (1–21 years) after illness onset, meaning some patients would not have had contact with the clinic for a long time when they were sent the questionnaire, so it is likely that both disease status and time since illness influenced outcome data. Of the 489 patients who were sent baseline questionnaires, 74% returned a follow-up questionnaire on at least one occasion (range 1–7). For the sample of 366 without baseline data for anxiety and depression, 76% returned a follow-up questionnaire on one occasion, while only 8% returned a questionnaire on more than one occasion. Outcome measures were also self-reported, and many participants did not complete all measures.

## Participant and study characteristics

The two studies identified in the updated search were: an RCT evaluating the 'Lightning Process' intervention alongside 'specialist medical care' compared with 'specialist medical care' alone[17]; and an observational cohort study assessing 'routine specialist care' over a 20-year period.[16] Studies from the previous reviews included the following. Four RCTs evaluating: inpatient programmes with predominantly behavioural approaches,[18 28] an online CBT programme[29 30] and intravenous gammaglobulin[27]; eight observational cohort studies evaluating: CBT,[19 25 32] CBT with pharmacotherapy,[24 31] an antiviral treatment[26] and an inpatient programme[23]; and two prospective observational community studies that did not assess a specified intervention.[21 22] Follow-up times varied from immediately post-treatment to 21 years. Total number of participants included across all studies was 965. Most sample sizes were small but ranged between one and 418. Participant ages ranged between 11 and 18. Most studies were conducted across Europe (UK, Netherlands, Spain) and Australia. One was in Japan, one in the USA (table 2).

None of the studies identified were specifically aimed at treating anxiety or depression in children with CFS/ME (all primary outcomes were measures of fatigue or recovery). Anxiety and/or depression were measured as secondary outcomes using a variety of self-report questionnaires including the Hospital Anxiety and Depression Scale (HADS),[33] Spence Children's Anxiety Scale (SCAS),[34] the State-Trait Anxiety Inventory for Children,[35] the Multidimensional Anxiety Scale for Children,[36] Spielberger State Trait Anxiety Questionnaire,[37] Beck Depression Inventory,[38] Children's Depression

**Table 2** Participant and study characteristics

| Author (year), country | Anxiety, depression or both? | Study design | Setting | Sample size | | Mean age, years | | Gender, female % | | CFS/ME diagnostic criteria | Primary outcome | Measure of anxiety/depression | Treatment specifically targeted to anxiety or depression? | Outcomes stratified by those with anxiety/depression? | Intervention | Control | Length of follow-up |
|---|---|---|---|---|---|---|---|---|---|---|---|---|---|---|---|---|---|
| | | | | Control | Intervention/case | Control | Intervention/case | Control | Intervention/case | | | | | | | | |
| **(A) Studies identified in updated review** | | | | | | | | | | | | | | | | | |
| Rowe (2019), Australia[16] | Both | Observational retrospective | Outpatient secondary care | N/A | 418 (789 recruited but 366 did not have baseline questionnaire) | N/A | 14.8 | N/A | 77% | CDC/Fukuda et al | Reported recovery‡ and duration of illness | STAI, BDI | No | No | Routine specialist medical care provided in the outpatient clinic. Described as a person-centred goal-oriented holistic programme which targets educational, physical, social and emotional aspects of life. | N/A | Mean: 8 years; range 1–21 years |
| Crawley (2018), UK[17] | Both | RCT | Outpatient secondary care | 49 | 51 | 14.5 | 14.7 | 78% | 75% | NICE | SF-36 PFS at 6 months | SCAS, HADS | No | No | Specialist medical care (Based on NICE guidance)+Lightning Process (3×4 hour sessions on consecutive days with groups of 2–5 young people. Theory sessions teach the stress response, how the mind and body interact and how thought processes can be either helpful or negative. Practical sessions involve participants identifying a goal (eg, stand up for longer) and are given cognitive strategies.) | Specialist medical care only | 3, 6, 12 months |
| **(B) Studies identified in previous reviews** | | | | | | | | | | | | | | | | | |
| Henderson (2014), USA[26] | Depression | Observational, retrospective, case-series | Outpatient secondary care | N/A | 15 (14 at follow-up) | N/A | 15.46 | N/A | 73% | CDC/Fukuda et al | Fatigue self-assessment scores (CFSI, FSS, FSI, MFSI) | CDI | No | Yes | Valacyclovir (antiviral) medication, initially 500 mg Twice daily, increasing after 2–3 weeks. Duration of treatment ranged from 3 to 60 months (mean 27.9 months). | N/A | Varied post-treatment |
| Rimes (2014), UK[32] | Anxiety | Observational case-control | Outpatient secondary care | 36 healthy controls | 49 (24 at follow-up) | 15 | 14.9 | 58% | 63% | CDC/Fukuda et al Oxford/Sharpe et al | School attendance | SCAS | No | No | CBT via telephone based guided self-help. 6 fortnightly sessions, 30 mins duration | N/A | 6 months |
| Nijhof (2012, 2013), Netherlands[29,30] | Anxiety | Both RCTs | Outpatient secondary care | 67 (63 at follow-up) | 68 (64 at follow-up) | 15.8 | 15.9 | 85% | 79% | CDC/Fukuda et al | School attendance, absence of severe fatigue and normal physical functioning | STAIC | No | No | Internet delivered CBT consisting of psychoeducation and 21 modules, with parallel child and parent sessions. FITNET therapist individually tailored intervention and initially responded to emails weekly, decreasing to fortnightly. Mean treatment duration 26.2 weeks (SD 7.3). | Treatment as usual including CBT (66%), rehabilitation treatment (22%), physical treatment (mostly graded exercise therapy; (49%), or alternative treatment (24%) | 2.5 years |
| Lloyd (2012), UK[25] | Both | Observational | Outpatient secondary care | N/A | 63 (52 at follow-up) | N/A | Median 15 | N/A | 63% | Oxford/Sharpe et al | Fatigue (Chalder Fatigue Questionnaire Total) and school attendance | SCAS, Birleson Depression Scale | No | No | CBT via telephone based guided self-help. 6 fortnightly sessions, 30 mins duration | N/A | 6 months |
| Kawatani (2011), Japan[24] | Depression | Observational | Outpatient secondary care | N/A | 19 | N/A | 13.6 | N/A | 63% | Jason et al[62] | Chalder's Fatigue Scale | Zung self-rating depression scale | No | No | CBT (average of 5 sessions over 6 months) and pharmacotherapy (antidepressants, antihypotensives, hypnotic agents) | N/A | 6 months |

Continued

**Table 2** Continued

| Author (year), country | Anxiety, depression or both? | Study design | Setting | Sample size | | Mean age, years | | Gender, female % | | CFS/ME diagnostic criteria | Primary outcome | Measure of anxiety/depression | Treatment specifically targeted to anxiety or depression? | Outcomes stratified by those with anxiety/depression? | Intervention | Control | Length of follow-up |
|---|---|---|---|---|---|---|---|---|---|---|---|---|---|---|---|---|---|
| | | | | Control | Intervention /case | Control | Intervention /case | Control | Intervention /case | | | | | | | | |
| Gordon (2010), Australia[18] | Depression | RCT | Inpatient secondary care | Aerobic group: 11 | Resistance group: 11 | Aerobic group: 16.2 | Resistance group: 15.6 | Not reported | | CDC/Fukuda et al | Exercise tolerance (time to fatigue) | BDI | No | No | 4 week inpatient programme including graded exercise therapy, psychological/psychiatric support, attendance at school. Patients randomised to either graded aerobic exercise training or progressive resistance training programme for 5 days/week for 4 weeks. The graded aerobic training consisted of 20–40 min of stationary cycling and treadmill exercise. The progressive resistance training involved 16 exercises performed with single set, moderate load and high repetitions. | N/A | Post-treatment |
| Gordon (2009), Australia[23] | Depression | Observational | Inpatient secondary care | N/A | 16 | N/A | 16 | Not reported | | CDC/Fukuda et al | Physical and physiological measures for example, aerobic capacity ($VO_2$ peak), time to fatigue, physical component score of SF-36 | BDI | No | No | 4 week inpatient programme including graded exercise therapy, psychological/psychiatric support, attendance at school, recreation and leisure intervention. | N/A | Post-treatment |
| Diaz Caneja (2007), Spain[31] | Anxiety | Observational case study | Outpatient secondary care | N/A | 1 | N/A | 15 | N/A | 100% | Oxford/Sharpe et al | Self-reported fatigue, pain symptoms | MASC | No | No | CBT+fluoxetine (initially 10 mg daily, increased after 1 week to 20 mg) | N/A | 3 months |
| Rimes (2007), UK[21] | Both | Observational prospective | Community | N/A | 1 case of CFS at time 1; 4 cases of CFS at time 2 | N/A | 13 | Not reported | | CDC/Fukuda et al | Incidence and prevalence of fatigue, chronic fatigue and CFS | DAWBA | No | No | None specifically stated or evaluated | N/A | 4–6 months |
| Van de Putte (2007), Netherlands[22] | Both | Observational prospective | Community | N/A | 40 at baseline, 36 at follow-up | N/A | 16 | N/A | 78% | CDC/Fukuda et al | Fatigue | SSTAQ, CDI | No | No | None specifically stated or evaluated | N/A | 18 months |
| Wright (2005), UK[28] | Anxiety | RCT | Outpatient secondary care | 6 (5 at follow-up) | 7 (6 at follow-up) | 12.9 | | 66% | 57% | Oxford/Sharpe et al | Global Health on Child Health Questionnaire | HADS | No | No | STAIRway to Health intervention is a structured rehabilitation programme including conceptualising CFS as having both physical and psychological components, formulating and addressing vicious cycles around activity, sleep, social isolation, physical deconditioning, and developing adaptive coping strategies while challenging negative and unhelpful attributions about illness and the future. | Pacing - focuses on limiting activity to the changing needs and responses of the body by avoiding overexertion and managing energy within an overall limit | 1 year |
| Denborough (2003), Australia[20] | Depression | Observational | Inpatient secondary care | N/A | 39 (19 at follow-up) | N/A | 16.2 | N/A | 90% | CDC/Fukuda et al | Global assessment of functioning, Chronic Fatigue Illness Disability Scale, FSS | BDI | No | No | 4 week inpatient programme, focused on graded exercise using hydrotherapy and physiotherapy. | N/A | 6 months |
| Chalder (2002), UK[19] | Both | Observational | Outpatient secondary care | N/A | 23 | N/A | 14.5 | N/A | 87% | Oxford/Sharpe et al | The fatigue questionnaire, school attendance | HADS | No | No | CBT based rehabilitation programme. Up to 15 sessions, 1 hour duration. | N/A | 6 months |
| Rowe (1997), Australia[27] | Anxiety | RCT | Outpatient secondary care | 35 | 36 | 15.6 | 15.3 | 75% | 58% | CDC/Fukuda et al | Functional score including school attendance, school work, social activity and physical activity | SSTAQ | No | No | 3 monthly infusions of gammaglobulin | 3 monthly infusions of placebo | 3 and 6 months |

CDC classification criteria for CFS/ME, also known as Fukuda criteria. Oxford criteria, also known as Sharpe et al criteria.
Global rating was measured on multiple scales of functioning (including school/work, stamina, recovery, social and symptomatology) from 1 to 10, with 10 being 'back to normal'.
†reported recovery was based on the question 'Do you feel you are no longer suffering from CFS?' (yes/no).
BDI, Beck's Depression Inventory; CBT, cognitive–behavioural therapy; CDC, Centers for Disease Control and Prevention; CDI, Children's Depression Inventory; CFS/ME, chronic fatigue syndrome/myalgic encephalomyelitis; DAWBA, Development and Well-Being Assessment; FSI, fatigue symptom inventory; FSS, fatigue severity scale; HADS, hospital anxiety and depression scale; MASC, multidimensional anxiety scale for children; MFSI, multidimensional fatigue symptom inventory–short form; NA, not available; NICE, National Institute for Health and Care Excellence; PFS, Physical Function Subscale; RCT, randomised controlled trial; SCAS, Spence Children's Anxiety Scale; SF-36, Short-Form-36; SSTAQ, Spielberger State-Trait Anxiety Questionnaire; STAIQ, state-trait anxiety inventory (for children).

Inventory,[39] the Birleson Depression Scale[40] and Zung's Self-rating depression scale.[41] One study used a diagnostic interview, the Development and Well-Being Assessment.[42] Six studies (including the two identified in the updated review) measured both anxiety and depression; five measured depression only; and five anxiety only (table 2).

## Treatment approaches and their efficacy treating anxiety and/or depression in paediatric CFS/ME

Of the 16 studies: one study evaluated routine specialist outpatient care[16]; one evaluated the Lightening Process outpatient intervention[17]; one evaluated the 'STAIRway to health' outpatient intervention[28]; six evaluated various outpatient CBT programmes[19 24 25 29–32]; two evaluated outpatient pharmacological interventions (antivirals[26] and gammaglobulins[27]); three evaluated inpatient programmes focused on graded exercise therapy[18 20 23]; and two were epidemiological observational studies so were uninformative about interventions.[21 22]

There were common cognitive and behavioural elements across the behavioural and CBT programmes, including: behavioural strategies for a goal-oriented graded approach to increasing activity, often with the goal to return to full-time education or to commit to a regular activity; cognitive strategies to address the psychological implications of CFS/ME, illness-related beliefs and negative thoughts; and psychoeducation about the consequence of the illness and tools to navigate this. They varied in their intensity (eg, inpatient treatment, consecutive daily 4-hour outpatient sessions, and fortnightly 30 min phone calls), duration of treatment (days to years), and modality (eg, face to face, telephone and online). The antiviral and gammaglobulin studies did not include these elements and were distinct from the other studies in their approach.

Table 3 summarises outcomes of depression and/or anxiety and other relevant findings for each included study from (1) the updated review and (2) previous reviews. Below, we discuss the efficacy of the treatment approaches in the 14 studies which evaluated an intervention, by whether they were (1) an outpatient or (2) an inpatient programme.

### Outpatient programmes

The two new studies from this updated review evaluated two outpatient programmes. Crawley *et al*[17] compared adding the Lightening Process intervention (https://lightningprocess.com) to specialist care (recommended by National Institute for Health and Care Excellence (NICE)[1]), to specialist medical care alone. The Lightening Process is developed from osteopathy, life coaching and neurolinguistic programming and more than 250 children use it for their CFS/ME each year in the UK.[43] It is delivered in intensive three, 4-hour sessions on consecutive days in small groups, with theory elements on the stress response, how the mind and body interact and how thought processes and language can be either helpful or negative, followed by practical sessions where participants

identify an activity goal and are given cognitive strategies to attempt it. The study showed a significant reduction in adjusted difference in mean depressive and anxiety symptoms at 12 months (−1.8, p=0.04 for depression; −14.5, p<0.001 for anxiety) among participants allocated to the Lightening Process intervention (in addition to specialist medical care) arm than those allocated to the specialist medical care-only control. The Lightening Process was more effective than specialist medical care at reducing anxiety symptoms compared with depression (at both 6 and 12 months follow-up). Outcomes in this study were not stratified by those with depression or anxiety, so we cannot comment on other CFS/ME outcomes (such as fatigue or recovery) in context of comorbid depression or anxiety.

The other study identified in this updated review evaluated routine specialist care delivered at the authors' CFS/ME outpatient clinic in Australia.[16] Routine specialist care offers a 'person-centred goal-oriented holistic programme' to 'target educational, physical, social and emotional aspects of life'. This includes symptom management (eg, sleep, migraine, dizziness, nausea, orthostatic intolerance, concentration difficulties) and focussing on increasing activity and a commitment to something enjoyable outside the home on a regular basis. This study measured depressive and anxiety symptoms at baseline but not post-treatment, so we cannot comment on the effectiveness of the intervention at reducing depression or anxiety. Instead, the study compared mean baseline depression and anxiety scores between those who had self-reported 'recovery', defined as answering 'yes' to the question 'Do you feel you are no longer suffering from CFS?' measured at a mean length of follow-up of 8 years (range 1–21). There was no difference in depression or anxiety at baseline between those who reported that they had recovered and those who had not that is, depression nor anxiety were found to be associated with recovery.

As per our previous reviews,[10 11] several studies have evaluated other outpatient programmes. Outpatient CBT interventions demonstrated inconsistent efficacy and varied in terms of delivery modality (family-focused; face to face; telephone or internet-delivered modules with therapist e-consults), intensity (15 weekly, hourly therapist-led sessions; six fortnightly 30 min telephone calls), duration of treatment (12 weeks to 1 year), and whether pharmacotherapy was offered alongside CBT (antidepressants and antihypotensives). Three observational studies showed that face-to-face and telephone CBT resulted in improved depression, anxiety, functioning and social adjustment.[19 25 32] An RCT showed that participants who received internet-based CBT demonstrated improvement in fatigue and school attendance at 6 months follow-up, compared with participants who received usual care.[30] However, the study did not measure anxiety at follow-up. Two studies that evaluated CBT alongside pharmacotherapy were uninformative as they either did not reassess mood at follow-up,[24] or reported on only a single case-study.[31] In terms of behavioural approaches,

**Table 3** Summary of outcomes for symptoms of depression and anxiety and other relevant findings for included studies

| Study | Measure of depression and anxiety | Pre treatment: depression, mean (SD) | | Pre treatment: anxiety, mean (SD) | | Post treatment: depression, mean (SD) | | Post treatment: anxiety, mean (SD) | | Statistical analysis of change in depression/anxiety symptomatology | | Summary of other relevant findings |
|---|---|---|---|---|---|---|---|---|---|---|---|---|
| | | Intervention /case | Control | Intervention /case | Control | Intervention /case | Control | Intervention /case | Control | Depression | Anxiety | |
| **(A) Studies identified in updated review** | | | | | | | | | | | | |
| Rowe (2019)[16] | BDI* (depression scale), STAI* (anxiety scale) | 13.8 (8.9) | N/A | 88.9 (24.9) | N/A | N/A | N/A | N/A | N/A | No statistical change because post-treatment scores were not measured. Instead, mean baseline depression and anxiety scores were compared between those who reported recovery‡ and those who did not, using the Student's t-test. | | Overall, 46.5% reported recovery; participants who were followed for >10 years, 68% reported recovery. Mean duration of illness was 5 years |
| Crawley et al (2018)[17] | HADS* (depression and anxiety scales), SCAS* (anxiety scale) | 7.5 (3.1) | 8.1 (4.4) | HADS: 8.8 (4.5), SCAS: 29.8 (16.9) | HADS: 10.4 (4.4), SCAS: 40.3 (20.1) | 6 months: 4.2 12 months: 2.8 | 6 months: 5.9 12 months: 4.6 | HADS 6 months: 6.1 12 months: 5.3 SCAS 6 months: 24.7 12 months: 19.6 | HADS 6 months: 9.7 12 months: 8.3 SCAS 6 months: 37.4 12 months: 36.3 | Adjusted difference in means† (95% CI, p value): 6 months: −1.5 (−3.5 to 0.5, 0.1) 12 months: −1.8 (−3.4 to −0.1, 0.04) | Adjusted difference in means† (95% CI, p value): HADS at 6 months: −3.5 (−5.6 to −1.5, 0.001) SCAS at 6 months: −10.0 (−18.5 to −1.5, 0.02) HADS at 12 months: −2.6 (−4.7 to −0.4, 0.019); SCAS at 12 months: 14.5 (−22.4 to −6.7, <0.001) | At 6 months, participants allocated to LP in addition to SMC (intervention) had better physical function and fatigue at than those allocated to SMC (control). At 12 months, participants allocated to LP in addition to SMC (intervention) had better fatigue and school attendance than those in SMC (control). Adding LP to SMC is cost-effective. |
| **(B) Studies identified in previous reviews** | | | | | | | | | | | | |
| Henderson (2014)[26] | CDI | 14 (2.83) 4 patients with mood disorder:16.8 (1.92) 11 patients without mood disorder: 12.73 (2.00) | N/A | N/A | N/A | Not reported | N/A | N/A | N/A | Not reported | N/A | All patients reported at least 80% self-rated improvement. Significant reduction in FSS, MSFI (all subscales). |
| Rimes et al (2014)[32] | SCAS | N/A | N/A | Cases: 22 (17) Median 16.0 (IQR 9.0–34.0) | Controls: Median 16.5 (IQR 8.0–22.8) | N/A | N/A | Not reported | N/A | N/A | T value (21)=2.1. p=0.005 | Adolescents with CFS had reduced cortisol excretion throughout the day compared with healthy controls. There was significant improvement in school attendance after treatment from 24% to 49%. There was reduction in fatigue after treatment, however the results were not significant. |

Continued

**Table 3** Continued

| Study | Measure of depression and anxiety | Pre treatment: depression, mean (SD) | | Pre treatment: anxiety, mean (SD) | | Post treatment: depression, mean (SD) | | Post treatment: anxiety, mean (SD) | | Statistical analysis of change in depression/anxiety symptomatology | | Summary of other relevant findings |
|---|---|---|---|---|---|---|---|---|---|---|---|---|
| | | Intervention /case | Control | Intervention /case | Control | Intervention /case | Control | Intervention /case | Control | Depression | Anxiety | |
| Nijhof et al (2012, 2013)[29 30] | STAIC | N/A | N/A | 32.7 (8.8) | 32.3 (8.0) | N/A | N/A | Not reported | N/A | N/A | Not reported | Intervention (FITNET) was significantly more effective than the control (usual care) at 6 months—full school attendance (50 (75%) vs 10 (16%), relative risk 4.8, 95% CI 2.7–8.9; p<0.0001), absence of severe fatigue (57 (85%) vs 17 (27%), 3.2, 2.1–4.9; p<0.0001), and normal physical functioning (52 (78%) vs 13 (20%), 3.8, 2.3–6.3; p<0.0001). The short-term effectiveness of FITNET was maintained at 2.5 years follow-up. At 2.5 years follow-up, usual care led to similar recovery rates, although progress had taken longer to make. At 6 months additional analyses of main findings with adjustments for anxiety, depression and primary outcomes, had no effects on the results. When looking at factors related to recovery at 2.5 years, anxiety OR 1.01 (95% CI 0.96 to 1.06), p=0.66 |
| Lloyd et al (2012)[25] | Birleson Depression Scale; SCAS | Baseline mean 13.38 (4.76) Pre-treatment mean 12.91 (5.57) | N/A | Baseline mean 22.84 (17.18) Baseline median 16.0 (IQR 10.8–35.0) | N/A | Post-treatment: 10.98 (5.35) 3 months: 10.47 (5.87) 6 months: 9.22 (5.36) | N/A | 6 months: 17.25 (3.06) | N/A | Multi-level modelling and Wald tests Treatment effect estimate at 6 months: 3.69 (95% CI –5.17 to –2.21), significance (two-tailed) <0.001, effect size 0.78. | Multi-level modelling and Wald tests Treatment effect estimate at 6 months: 0.49, significance (two-tailed) 0.003, effect size 0.16 | Significant improvement in fatigue and school attendance, with reductions in depression and impairment and increased adjustment at 6 months |
| Kawatani et al (2011)[24] | Zung Self-Rating Depression Scale | 53.3 (6.7) | N/A | N/A | N/A | Not reported | N/A | N/A | N/A | Not reported | N/A | No significant change between baseline fatigue scores and fatigue scores 6 months follow-up. Significant improvement in performance status scores (self-reported impact on functioning). |
| Gordon et al (2010)[18] | BDI | Resistance arm: 20.9 (11.3) | Aerobic arm: 16.4 (4.3) | N/A | N/A | Resistance arm: 14.2 (10.0) | Aerobic arm: 12.2 (6.7) | N/A | N/A | Resistance arm Difference –6.7±8.5 p=0.03 Aerobic arm Difference –4.2±4.8 p=0.002 | N/A | There was no control group. Significant improvement in BDI scores in both arms. |
| Gordon and Lubitz (2009)[23] | BDI | 19.88 (8.62) | N/A | N/A | N/A | 11.44 (10.98) | N/A | N/A | N/A | Paired t-test p value 0.001, sig 0.008 | N/A | Significant improvement in Fatigue Severity scores. |

Continued

**Table 3** Continued

| Study | Measure of depression and anxiety | Pre treatment: depression, mean (SD) | | Pre treatment: anxiety, mean (SD) | | Post treatment: depression, mean (SD) | | Post treatment: anxiety, mean (SD) | | Statistical analysis of change in depression/anxiety symptomatology | | Summary of other relevant findings |
|---|---|---|---|---|---|---|---|---|---|---|---|---|
| | | Intervention /case | Control | Intervention /case | Control | Intervention /case | Control | Intervention /case | Control | Depression | Anxiety | |
| Diaz-Caneja et al (2007)[31] | MASC | N/A | N/A | Not stated. Raised levels of social anxiety and physical symptoms of anxiety | N/A | N/A | N/A | Not stated although it is reported that anxiety improved | N/A | N/A | Not reported | Report of a moderate response to treatment with the young person tolerating more activity. She had resumed contact with her friends, and although she still complained of tiredness and pain, she was attending classes daily. |
| Rimes et al (2007)[21] | DAWBA | Only states '3 of 4 had at least one psychiatric diagnosis at baseline' | N/A | Only states '3 of 4 had at least one psychiatric diagnosis at baseline' | N/A | N/A | N/A | N/A | N/A | Not reported | Not reported | Of the four participants who developed CFS/ME over the follow-up period, 3 of 4 had at least one psychiatric diagnosis at baseline, 3 had reported being 'much more tired and worn out than usual over the last month' at time 1, 2 participants had frequent headaches at time 1, 1 also had sleep problems and post-exertional malaise at time 1. |
| Van de Putte et al (2007)[22] | CDI at baseline only; HADS (anxiety) | 11.7 (6.1) | N/A | 36.9 (7.8) | N/A | Not stated | N/A | Not stated | N/A | Not reported | Not reported | 47% of adolescents 'fully recovered' (below score that is mean plus 2 SD of subjective fatigue distribution in health adolescents). |
| Wright et al (2005)[28] | HADS (anxiety) | N/A | N/A | 10.17 (3.71) | 6.80 (3.56) | N/A | N/A | Post-treatment: 6.00 (3.63) | Post-treatment: 6.60 (4.73) | N/A | Analysis of covariance for anxiety, controlling for baseline score. Difference −1.60 (−8.31–5.10) F 0.3 (df 1,8) p=0.6 | Activity (child and clinician rated) and school attendance improved markedly in the intervention (STAIRway) arm compared with little improvement in activity scores in the control (Pacing) arm, and a deterioration in school attendance. Global health (child and clinician rated) improved in both arms although more in the STAIRway arm than the pacing arm. |
| Denborough et al (2003)[20] | BDI | 21 | N/A | N/A | N/A | 15 | N/A | N/A | N/A | Improvement p<0.001 Maintained at 6 month follow-up (p<0.038) | N/A | On discharge, mean depression score significantly better than on admission. Also significant improvement in Chronic Fatigue Illness Disability score and significant decrease in FSS score (maintained at 6 months follow-up). Achenbach/Youth Self-Report scores improved significantly by discharge, but returned to above admission levels at 6 months. |

Continued

**Table 3** Continued

| Study | Measure of depression and anxiety | Pre treatment: depression, mean (SD) | | Pre treatment: anxiety, mean (SD) | | Post treatment: depression, mean (SD) | | Post treatment: anxiety, mean (SD) | | Statistical analysis of change in depression/anxiety symptomatology | | Summary of other relevant findings |
|---|---|---|---|---|---|---|---|---|---|---|---|---|
| | | Intervention /case | Control | Intervention /case | Control | Intervention /case | Control | Intervention /case | Control | Depression | Anxiety | |
| Chalder et al (2002)[19] | HADS | 8.4 (IQR 5.7–11) | N/A | HADS anxiety: median 7, (IQR 6.7–9.7) | N/A | 6 months: 3 (IQR 3–5) | N/A | 6 months: HADS anxiety: 0.5 (IQ range 0.5–9) | N/A | Wilcoxon signed ranks test −3.33 (2 tailed significance 0.00) | Wilcoxon signed ranks test (significance two tailed) HADS anxiety: 2.02 (0.04) | Depression: The 20 participants who completed treatment had all returned to school at 6 months follow-up, with 19 of 20 attending full time. Depression significantly improved, as did social adjustment. Anxiety: All 20 treatment completers returned to school at 6 months follow-up, with 95% attending full time. Depression significantly improved, as did social adjustment. |
| Rowe (1997)[27] | SSTAQ | N/A | N/A | Reported as one group: Mean 46.2 (24.4) SE 3.9 Range 0–98 | N/A | N/A | N/A | 6 months: Mean 28.1 (25.0) SE 5.9 Range 0–77 | | N/A | T value (df) 2.63 (56) Sig p value 0.01 | Significant mean functional improvement in both groups. |

*Higher score=more symptoms, poorer function.
†Adjusted for age, gender, baseline outcome, SCAS and Visual Analogue Scale.
‡Reported recovery was based on the question: 'Do you feel you are no longer suffering from CFS?' (yes/no)
BDI, Beck's Depression Inventory; CDI, Children's Depression Inventory; CFS/ME, chronic fatigue syndrome/myalgic encephalomyelitis; DAWBA, Development and Well-Being Assessment; FSI, Fatigue Symptom Inventory; FSS, Fatigue Severity Scale; HADS, Hospital Anxiety and Depression Scale; LP, lightning process; MASC, Multidimensional Anxiety Scale for Children; MFSI, Multidimensional Fatigue Symptom Inventory–Short Form; N/A, not available; SCAS, Spence Children's Anxiety Scale; SMC, Specialist Medical Care; SSTAQ, Spielberger State-Trait Anxiety Questionnaire; STAI(C), State-Trait Anxiety Inventory (for children).

the STAIRway to Health—an incremental rehabilitation intervention—showed greater improvement in anxiety levels, when compared with a 'pacing' intervention in an RCT.[28] Pharmacological studies showed insufficient evidence for improving anxiety or depressive symptoms with intravenous gammaglobulin infusions or vancyclovir respectively[26 27]

## Inpatient programmes

As per our previous review,[10] three studies[18 20 23] including one RCT, evidenced an improvement in mood post-treatment with a 4-week inpatient behavioural programme focused on graded exercise (including physiotherapy, aerobic exercise and resistance training), which were maintained at 6 month follow-up in one study[20]). However, they did not measure anxiety symptoms; internalising problems at 6 months returned to preadmission levels; two studies did not have follow-up data[18 23]; all studies had small sample sizes; and the multicomponent intervention also included psychological therapy (with no further specified details about this). Therefore, these studies are uninformative for drawing conclusions about the efficacy of this behavioural intervention, or about what the key effective components of the approach may have been.

## DISCUSSION

Our updated review of interventions for comorbid depression and/or anxiety in children with CFS/ME identified only two new studies published since 2015 (one of which was conducted by members of our own research team) exposing the lack of progress in this field. One study (an RCT) showed that adding the Lightening Process intervention to specialist medical care was more effective than specialist medical care alone at reducing both depressive and, to a greater extent, anxiety symptoms. The other study (an observational cohort evaluating routine specialist care) did not measure depression or anxiety at follow-up. Combined with our results from previous reviews, we identified 16 studies of 11 different interventions for paediatric CFS/ME since 1991 that include measures of anxiety and/or depression. Of these, six did not provide follow-up measurements of anxiety and/or depression post-intervention, and none of the interventions in the studies specifically targeted comorbid anxiety and/or depression. The results of this updated review do not appreciably alter what is already known from previous reviews, that there is insufficient evidence to conclude what the best interventions are for treating anxiety and/or depression in paediatric CFS/ME patients.

Strengths of the updated review include the systematic approach, the use of four reviewers, contacting authors for subgroup data, and not limiting results to English language. The limitations are the lack of eligible studies and insufficient data available for a meta-analysis. Only two papers were eligible for inclusion, of which one did not provide sufficient follow-up data to comment on the treatment efficacy of the intervention on depression and

anxiety. Neither intervention was specifically designed to measure the impact on depression and anxiety and therefore studies were inadequately powered to measure this. Studies were not stratified by those who met criteria for clinical diagnoses of depression/anxiety reducing our ability to analyse effectiveness. Furthermore, neither study used diagnostic interviews for anxiety and depression, relying instead on questionnaires. While HADS,[44] SCAS[45] and STAI[35] questionnaires are validated for use in adolescents, only the RCADS (Revised Children's Anxiety and Depression scale), which is derived from the SCAS, has been found to have sufficient discriminative accuracy against gold standard diagnostic interviews in paediatric CFS/ME populations.[5]

In conjunction with our previous reviews, we show that currently the interventions with most evidence for improvement in anxiety and depressive symptoms in CFS/ME, when compared with other interventions, such as behavioural-only or pharmacological, is CBT.[10 11] The 'Lightening Process' programme, 'STAIRway to Health' intervention, and a 4-week multicomponent inpatient rehabilitation programme show promising results for improving anxiety and/or depressive symptoms in single RCTs, but sample sizes are small and results have not been replicated. The mechanisms for why CBT could be effective are unclear because no study targeted anxiety and depression. Further, multicomponent outpatient and inpatient interventions make it difficult to identify the effective element of interventions. Our updated review does not further this debate because, while CBT is an element of 'specialist medical care' and 'routine specialist care' interventions in the new studies, we do not know how many participants received CBT or how it was delivered. Additionally, results are not stratified by those with anxiety and/or depression. Furthermore, the differences and similarities between the Lightening Process and CBT are also unclear.[46] It should also be noted that the draft NICE guideline (expected publication date August 2021: https://www.nice.org.uk/guidance/gid-ng10091/documents/draft-guideline) does not recommend the Lightning Process for management of CFS (although this is not specifically aimed at anxiety and depression).

Other cognitive and behavioural based approaches are being trialled in CFS/ME, but are limited in contributing to our understanding of their efficacy for anxiety and depressive symptoms in CFS/ME because of a failure to include paediatric CFS/ME populations or those diagnosed with CFS/ME using recognised criteria, or measure anxiety and depressive symptoms in the 20%–30%[5 6] of children that experience them. Three studies[47–49] were excluded from our review for these reasons. For example, studies evaluating acceptance and commitment therapy[47] and mindfulness-based therapies[48] show promising results in improving the physical health, symptom burden and 'emotional distress' in children with functional somatic syndromes including CFS/ME but were excluded from this review because data for adolescent participants with CFS/ME were aggregated with those with other somatic

syndromes, and the studies only measured general well-being outcomes rather than specifically validated anxiety and/or depression outcomes.

There is a pressing need for more work in this area to identify efficacious treatments for anxiety and depressive symptoms in paediatric CFS/ME so they can be used in clinical practice. We call on researchers to undertake paediatric CFS/ME interventions studies and use validated, diagnostic outcome measures of anxiety and depression.

## CONCLUSION

This updated review highlights both the paucity of intervention studies in children with CFS/ME since 1991 and the lack of forward movement in identifying effective treatments for paediatric CFS/ME and comorbid depression and anxiety over the last five years. The overall quality of the literature remains poor and calls for paediatric CFS/ME intervention studies to target anxiety and depression, measure outcomes with validated scales, or report outcomes in subsets of patients with clinical diagnoses of anxiety and depression, have not been met. Given that comorbid anxiety and depression in paediatric CFS/ME are associated with worse outcomes, unlikely to remit spontaneously without treatment, and can be incompatible with following standard CFS/ME treatment guidance, this needs to be addressed. Future research should improve the quality of the literature by using validated scales (as well as analyse correlation between scales) and measure anxiety and/or depression as primary outcomes in large intervention studies of comorbid anxiety and/or depression in paediatric CFS/ME.

**Acknowledgements** We would like to acknowledge the support from the CFS/ME teams at the Centre for Academic Child Health at the University of Bristol and the CFS/ME service at the Royal United Hospitals Bath NHS Foundation Trust.

**Contributors** ML and EC (guarantor) conceptualised this study. PC, AR, KD and JB performed data collection, synthesis and interpretation. PC wrote the manuscript. All authors contributed to manuscript revisions, have read the final manuscript and approved it for publication. All authors agree to be accountable for all aspects of the work.

**Funding** ML is funded by the National Institute for Health Research (NIHR Doctoral Research Fellowship, DRF-2016-09-021). This report is independent research.

**Disclaimer** The views expressed in this publication are those of the authors and not necessarily those of the NHS, NIHR or the Department of Health and Social Care.

**Competing interests** EC acts as a non-paid medical advisor for the Sussex and Kent ME society.

**Patient consent for publication** Not applicable.

**Provenance and peer review** Not commissioned; externally peer reviewed.

**Data availability statement** Data sharing not applicable as no datasets generated and/or analysed for this study. Data sharing not applicable as no datasets were generated or analysed for this study.

**ORCID iD**
Philippa Clery http://orcid.org/0000-0002-6770-4454

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
