## [Reviewer comments · BMJ Open]

ARTICLE DETAILS

TITLE (PROVISIONAL)	What treatments work for anxiety and depression in children and adolescents with Chronic Fatigue Syndrome? An updated systematic review
AUTHORS	Clery, Philippa; Royston, Alexander; Driver, Katie; Bailey, Jasmine; Crawley, Esther; Loades, Maria

VERSION 1 – REVIEW

REVIEWER	Thabrew, Hiran University of Auckland, Psychological Medicine
REVIEW RETURNED	05-May-2021

GENERAL COMMENTS	Thank you for the opportunity to review this interesting update of two previous reviews of anxiety and depression changing interventions in children with CFS/ME. While well-written and performed to a high standard by an experienced team of researchers, the small number of included studies is disappointing and not clinically very useful to the reader. Only two studies were actually analysed and one of these (Rowe 2019) did not include follow-up measurement of anxiety or depression, so would not seem to meet the eligibility criteria provided by the authors in table 1. The other study (Crawley 2018) was carried out by one of the review team. I do not think the results of this review need to be wasted. I would suggest combining them with those of the previous two reviews to provide readers with a total view of the number and types of interventions that have been evaluated in this clinical cohort, and possibly a meta-analysis of findings from this larger number of studies (I note around 9 types of intervention measuring changes in anxiety and 9 types of intervention measuring changes in depression were included in the previous reviews). I sincerely believe clinicians working with this group of children would appreciate such an overview and summary of evaluated interventions.

REVIEWER	Quinn, Steve Flinders, Biostats
REVIEW RETURNED	10-May-2021

GENERAL COMMENTS	This paper provides an update on studies conducted in children with CFS/ME and the subsequent impact on depression and anxiety, from 2015-2020 inclusive. Two new studies, one RCT and one observational cohort study were identified, neither of which specifically targeted anxiety and/or depression. The search
---

	methods were rigorous and considerable space was devoted to bias. Details of the conduct and design of both studies are provided, probably more than the reader would expect in a systematic review. Whilst the study is scientifically sound, the take-home message is that there were not enough new studies to conduct a meta-analysis. This might explain why the conduct and design of both studies are provided in great detail, probably more than the reader would expect in a systematic review where several candidate studies are identified. Presumably the results of the two studies did not appreciably alter what is already known in previous meta-analyses, but this is not commented upon in the discussion. It is important for interested readers to learn of what is happening in this area, but at the end of the day the authors have very little to report that adds to the body of knowledge or will change clinical practice. It may be more appropriate for the authors to report these results and findings in letter format, given the dearth of new studies.
--	--

REVIEWER	YAO, FEI
REVIEW RETURNED	23-May-2021

GENERAL COMMENTS	The topic selection has certain practical significance and certain guidance. It is necessary to further discuss how the intervention method can improve anxiety and depression. References need further update The overall quality of the literature is poor and needs further screening The correlation between the scales needs further analysis
---

VERSION 1 – AUTHOR RESPONSE

Reviewer: 1 Dr. Hiran Thabrew, University of Auckland	Our Response
Thank you for the opportunity to review this interesting update of two previous reviews of anxiety and depression changing interventions in children with CFS/ME. While well-written and performed to a high standard by an experienced team of researchers, the small number of included studies is disappointing and not clinically very useful to the reader. Only two studies were actually analysed and one of these (Rowe 2019) did not include follow-up measurement of anxiety or depression, so would not seem to meet the eligibility criteria provided by the authors in table 1. The other study (Crawley 2018) was carried out by one of the review team.	Thank you for reviewing our update and providing comments. Regarding your comment that the Rowe 2019 paper didn't meet eligibility criteria (Table 1). The Table states that the outcome measure should be a repeated measure of either anxiety/depression and/or fatigue. The Rowe paper meets this criteria for fatigue. We appreciate this is not particularly clear and so have changed the wording slightly and used bold typeface to make it more clear. It now reads: Either an anxiety and/or fatigue measure on psychometrically validated assessments or validated diagnostic interviews

	Either a depression and/or fatigue measure on psychometrically validated assessments or validated diagnostic interviews.
I do not think the results of this review need to be wasted. I would suggest combining them with those of the previous two reviews to provide readers with a total view of the number and types of interventions that have been evaluated in this clinical cohort, and possibly a meta-analysis of findings from this larger number of studies (I note around 9 types of intervention measuring changes in anxiety and 9 types of intervention measuring changes in depression were included in the previous reviews). I sincerely believe clinicians working with this group of children would appreciate such an overview and summary of evaluated interventions.	Thank you for your comment. We have now significantly altered the paper to combine the results from these two included studies, with findings from the two previous systematic reviews to provide an overview of all interventions that have been evaluated in this clinical cohort since 1991. This is done throughout the paper, but notably in the sections highlighted as above in the comment to the associate editor (please refer to this). Thank you for your suggestion of a meta-analysis. However, it is regrettable that given the heterogeneous interventions, measures, and follow-up periods, there is insufficient comparable data to conduct a meta-analysis. Further, displaying results of a meta-analysis would suggest or may give the reader the idea that the data are combinable, or the studies are comparable, but a key finding we discuss is that there is little comparability and little standard across these interventions in this clinical cohort.

Reviewer 2: Dr. Steve Quinn, Flinders	Our Response
This paper provides an update on studies conducted in children with CFS/ME and the subsequent impact on depression and anxiety, from 2015-2020 inclusive. Two new studies, one RCT and one observational cohort study were identified, neither of which specifically targeted anxiety and/or depression. The search methods were rigorous and considerable space was devoted to bias. Details of the conduct and design of both studies are provided, probably more than the reader would expect in a systematic review.	Thank you for your comment.
Whilst the study is scientifically sound, the take-home message is that there were not enough new studies to conduct a meta-analysis. This might	Thank you for your comment. We have now significantly altered the manuscript to combine findings from previous reviews with the results of our updated review. We hope

explain why the conduct and design of both studies are provided in great detail, probably more than the reader would expect in a systematic review where several candidate studies are identified. Presumably the results of the two studies did not appreciably alter what is already known in previous meta-analyses, but this is not commented upon in the discussion.	this provides a more clear overview of what is known and what is new. We have updated the discussion to read: P22 line 23-4, P23 lines 1-2: The results of these two studies do not appreciably alter what is already known from previous reviews, that there is insufficient evidence to conclude what the best interventions are for treating anxiety and/or depression in paediatric CFS/ME patients.
It is important for interested readers to learn of what is happening in this area, but at the end of the day the authors have very little to report that adds to the body of knowledge or will change clinical practice. It may be more appropriate for the authors to report these results and findings in letter format, given the dearth of new studies.	Thank you. As mentioned, the manuscript now combines the findings from previous reviews in order to provide a more comprehensive overview. We therefore believe it worthy of an article, rather than a letter to the editor.

Reviewer 3: Dr. FEI YAO	
The topic selection has certain practical significance and certain guidance.	Thank you.
It is necessary to further discuss how the intervention method can improve anxiety and depression.	Thank you. We agree and write in the discussion that it is unclear what the mechanisms or key components of approaches are that result in an improvement in anxiety and depression. P24 lines 2-8: The mechanisms for why CBT could be effective are unclear because no study targeted anxiety and depression. Further, multi-component outpatient and inpatient interventions make it difficult to identify active interventions. Our updated review does not further this debate because, whilst CBT is often an element of 'specialist medical care' and 'routine specialist care' in the identified studies, we do not know how many participants received CBT or how it was delivered. Additionally, results are not stratified by those with anxiety and/or depression. Furthermore, the differences and similarities between the Lightning Process and CBT are also unclear[49].
References need further update	Thank you. We have updated the references as we have re-written the manuscript to include studies from previous reviews.

The overall quality of the literature is poor and needs further screening	Thank you. We agree and note this in our discussion and conclusions. P25 lines 9-18: The overall quality of the literature remains poor and calls for paediatric CFS/ME intervention studies to target anxiety and depression, measure outcomes with validated scales, or report outcomes in subsets of patients with clinical diagnoses of anxiety and depression, have not been met. Given that comorbid anxiety and depression in paediatric CFS/ME are associated with worse outcomes, unlikely to remit spontaneously without treatment, and can be incompatible with following standard CFS/ME treatment guidance, future research needs to: improve the quality of the literature by using validated scales (as well as analyse correlation between scales) and measure anxiety and/or depression as primary outcomes in large intervention studies of comorbid anxiety and/or depression in paediatric CFS/ME.
The correlation between the scales needs further analysis	Thank you. We agree and note this in our conclusion: P25 lines 9-18: The overall quality of the literature remains poor and calls for paediatric CFS/ME intervention studies to target anxiety and depression, measure outcomes with validated scales, or report outcomes in subsets of patients with clinical diagnoses of anxiety and depression, have not been met. Given that comorbid anxiety and depression in paediatric CFS/ME are associated with worse outcomes, unlikely to remit spontaneously without treatment, and can be incompatible with following standard CFS/ME treatment guidance, future research needs to: improve the quality of the literature by using validated scales (as well as analyse correlation between scales) and measure anxiety and/or depression as primary outcomes in large intervention studies of comorbid anxiety and/or depression in paediatric CFS/ME.